# Radiation-Induced Innate Neutrophil Response in Tumor Is Mediated by the CXCLs/CXCR2 Axis

**DOI:** 10.3390/cancers15235686

**Published:** 2023-12-01

**Authors:** Faya Zhang, Oscar Mulvaney, Erica Salcedo, Subrata Manna, James Z. Zhu, Tao Wang, Chul Ahn, Laurentiu M. Pop, Raquibul Hannan

**Affiliations:** 1Department of Radiation Oncology, University of Texas Southwestern Medical Center, Dallas, TX 75390, USA; faya.zhang@utsouthwestern.edu (F.Z.); oscar.mulvaney@utsouthwestern.edu (O.M.); erica.salcedo@utsouthwestern.edu (E.S.); subrata.manna@utsouthwestern.edu (S.M.); laurentiu.pop@utsouthwestern.edu (L.M.P.); 2Quantitative Biomedical Research Center, Department of Population and Data Sciences, University of Texas Southwestern Medical Center, Dallas, TX 75390, USA; james.zhu@utsouthwestern.edu (J.Z.Z.); tao.wang@utsouthwestern.edu (T.W.); 3Department of Population and Data Sciences, University of Texas Southwestern Medical Center, Dallas, TX 75390, USA; chul.ahn@utsouthwestern.edu

**Keywords:** CXCLs, CXCR2, radiation therapy, IL-1β, neutrophils, DNA sensing, tumor microenvironment

## Abstract

**Simple Summary:**

Neutrophilic infiltration into the tumor microenvironment (TME) in response to radiation therapy (RT) is an early innate inflammatory immune response that occurs within 24–48 h where the neutrophils exhibit an anti-tumor phenotype. It is postulated that RT-induced DNA damage leads to innate sensing of DNA activating the cGAS-STING pathway, which is responsible for the immune modulatory effects of RT. We report that the CXCLs/CXCR2 axis translates the downstream effect of RT-induced innate DNA sensing resulting in an early innate neutrophilic response in the TME. This study identifies CXCLs/CXCR2 as a potential therapeutic target to improve RT-induced innate immune response.

**Abstract:**

The early events that lead to the inflammatory and immune-modulatory effects of radiation therapy (RT) in the tumor microenvironment (TME) after its DNA damage response activating the innate DNA-sensing pathways are largely unknown. Neutrophilic infiltration into the TME in response to RT is an early innate inflammatory response that occurs within 24–48 h. Using two different syngeneic murine tumor models (RM-9 and MC-38), we demonstrated that CXCR2 blockade significantly reduced RT-induced neutrophilic infiltration. CXCR2 blockade showed the same effects on RT-induced tumor inhibition and host survival as direct neutrophil depletion. Neutrophils highly and preferentially expressed CXCR2 compared to other immune cells. Importantly, RT induced both gene and protein expression of CXCLs in the TME within 24 h, attracting neutrophils into the tumor. Expectedly, RT also upregulated the gene expression of both cGAS and AIM2 DNA-sensing pathways in cGAS-positive MC-38 tumors but not in cGAS-negative RM-9 tumors. Activation of these pathways resulted in increased IL-1β, which is known to activate the CXCLs/CXCR2 axis. Gene ontology analysis of mRNA-Seq supported these findings. Taken together, the findings suggest that the CXCLs/CXCR2 axis mediates the RT-induced innate inflammatory response in the TME, likely translating the effects of innate DNA-sensing pathways that are activated in response to RT-induced DNA damage.

## 1. Introduction

Radiation therapy (RT) is one of the pillars of cancer treatment, and about half of patients with cancer receive RT during the course of their illness [1,2]. RT induces DNA damage and has direct cytotoxic effects on tumor cells. The improper DNA damage repair of double-strand breaks can lead to apoptosis, mitotic catastrophe, or senescence of tumor cells [3]. Additionally, RT modulates the tumor microenvironment (TME) via multiple pathways, including shaping immunogenicity of tumor cells and cell death, promoting tumor antigen presentation leading to tumor-specific T-cell priming, and inducing leukocyte infiltration into the TME [4]. Therefore, a better understanding of how RT affects the TME and the immune system will facilitate the harnessing of the immune-modulatory properties of RT in designing successful RT and immunotherapy combination trials that improve cancer outcomes.

Neutrophils are the most abundant immune cell population in humans and mice, and they play important roles in wound healing, chronic inflammation, infection, and cancer [5]. Neutrophils exhibit plasticity and can exert both pro-tumor and anti-tumor functions. On the one hand, neutrophils can increase cancer genetic instability, promote tumor cell proliferation, facilitate tumor angiogenesis, and suppress anti-tumor immunity [6]. On the other hand, neutrophils can directly kill tumor cells through reactive oxygen species (ROS), promote anti-tumor adaptive immunity, limit microbiota-induced tumor-promoting inflammation, and activate anti-tumor TCRαβ^+^CD4^−^CD8^−^ double-negative unconventional T cells [6].

Using syngeneic mouse tumor models, we demonstrated that local RT induced sterile inflammation with a rapid and transient infiltration of neutrophils into the tumors at 24 h [7]. In corroboration with our finding, RT was also found to increase tumor-infiltrating neutrophils within three days post-treatment in other tumor models [8,9]. These RT-recruited tumor-associated neutrophils exhibited an increased production of ROS, induced apoptosis of tumor cells, and eventually activated tumor-specific cytotoxic T cells [7]. Concurrent administration of neutrophil stimulatory cytokine granulocyte colony-stimulating factor enhanced RT-mediated anti-tumor activity [7]. These results suggest that the anti-tumor effects of RT are partially mediated through recruiting neutrophils into the tumor. Moreover, it is well known that RT induces DNA damage, and it has been postulated that DNA damage can induce multiple DNA-sensing pathways in the TME, leading to increased downstream cytokines such as type-I interferon (IFN) [10,11,12] and interleukin-1 beta (IL-1β) [13]. However, the subsequent events that lead to the eventual recruitment of RT-induced infiltration of neutrophils are not yet known.

Neutrophil migration from the periphery to different organs or inflammatory foci and subsequent infiltration into the tissue requires a complex interplay between multiple types of molecules, including cytokines and chemokines. The concentration gradients of chemokines can attract neutrophils toward the sites of inflammation through surface expressions of the respective chemokine receptors on neutrophils. The subsequent infiltration to the tissue also involves the interaction between other neutrophil cell surface molecules (e.g., integrins) and adhesion molecules on the endothelial cells [14]. C-X-C motif chemokine receptor 2 (CXCR2) and C-X-C motif chemokine ligands (CXCLs) (e.g., CXCL1, CXCL2, CXCL5) are known to be involved in recruiting neutrophils to the inflamed tissues [14]. Furthermore, CXCR2 and CXCLs have been shown to be involved in the recruitment of neutrophils into the tumor [15]. Therefore, we hypothesized that the CXCLs/CXCR2 axis mediates RT-induced neutrophilic infiltration of the tumor.

Interleukin 1 beta (IL-1β) is one of the cytokines that could regulate gene expression of CXCLs [16]. IL-1β has been shown to induce CXCL1 and CXCL2 expression in vitro [17] and in vivo [18]. RT can induce IL-1β as early as 2 h post-treatment in keratinocytes [19], which mediates RT-induced skin injury [20]. Interestingly, RT induces IL-1β expression not only in bone marrow-derived macrophages [21] but also in a human lung cancer cell line [22]. Therefore, we hypothesized that IL-1β may be an upstream signaling molecule that translates the RT-induced DNA damage and DNA sensing by inflammasomes into RT-induced neutrophilic infiltration via the activation of the CXCLs/CXCR2 axis.

In this study, we used two different types of syngeneic mouse tumor models (prostate and colorectal) to examine if pharmacological blockade of CXCR2 could reduce RT-induced neutrophil recruitment into the tumor. We then examined CXCR2 expression on neutrophils and other immune cells in the tumor-bearing mice. Subsequently, we measured protein levels of CXCR2 ligands (CXCLs) in the TME at early time points post-RT. Furthermore, we performed an mRNA sequencing experiment to examine if RT increases the gene expression of DNA-sensing pathways and conducted a gene ontology analysis to examine if RT activates related biological processes. Finally, we examined if radiation induces the protein expression of IL-1β in the TME.

## 2. Materials and Methods

### 2.1. Animals

Male C57BL/6J mice between six and seven weeks of age were purchased from The Jackson Laboratory (Bar Harbor, ME). All mice were maintained under specific pathogen-free conditions within the facilities of the Animal Resource Center at the University of Texas Southwestern Medical Center at Dallas in Texas (UTSW), following the guidelines from the National Institutes of Health. All animal protocols were approved by the Institutional Animal Care and Use Committee at UTSW.

### 2.2. Cell Lines

Mouse prostate cancer cell line RM-9 was purchased from the American Type Culture Collection (ATCC). Mouse colon cancer cell line MC-38 was purchased from Kerafast, Inc. (Shirley, MA, USA). Both RM-9 and MC-38 cell lines were cultured in Dulbecco’s Modified Eagle’s Medium (DMEM) supplemented with 10% fetal bovine serum. Cell cultures were maintained at 37 °C in a humidified atmosphere containing 5% CO_2_. Both cell lines were routinely tested for mycoplasma contamination using the Universal Mycoplasma Detection Kit from ATCC (Cat# 30-1012K).

### 2.3. Antibodies

For flow cytometry staining, the following fluorophore-labeled anti-mouse monoclonal antibodies against cell surface proteins were used: FITC-CD45 (clone 30-F11), PE-F4/80 (clone RM8), PE-Cy7-CD11b (clone M1/70), APC-CD11c (clone N418), eFluor 450-CD11c (clone N418), PerCP-Cy5.5-Ly6C (clone HK1.4), APC-eFluor 780-Ly6G (clone 1A8), FITC-CD19 (clone eBio1D3), PE-CD25 (clone PC61.5), PE-Cy7-CD3 (clone 145-2C11), AF700-CD4 (clone GK1.5), PerCP-Cy5.5-CD8 (clone 53-6.7), and eFluor 450-CD49b (clone DX5), and these antibodies were purchased from Thermo Fisher Scientific (eBioscience™, Waltham, MA, USA); BV421-CXCR2 (clone V48-2310) was purchased from BD Biosciences (Franklin Lakes, NJ, USA); APC-CXCR2 (clone SA044G4) was purchased from BioLegend (San Diego, CA, USA). For the western blot experiment, anti-IL-1β antibody (EPR16805-15) was purchased from Abcam (Cambridge, UK); anti-cGAS antibody (clone D3O8O), anti-vinculin antibody (clone E1E9V), and horseradish peroxidase (HRP)-linked anti-rabbit IgG antibody were purchased from Cell Signaling Technology (Danvers, MA, USA).

### 2.4. Tumor Inoculation and Irradiation

The RM-9 cell line (1 × 10^6^ cells) or MC-38 cell line (1 × 10^6^ to 2 × 10^6^ cells) was injected subcutaneously (s.c.) into the right hind legs of mice. When tumors were palpable, tumor sizes were measured by a caliper three times a week. Once the tumor size reached about 8 mm in diameter at around 10 days after inoculation, tumors were locally irradiated with a single dose of 15 Gy using an X-RAD 320 irradiator (Precision X-Ray). The irradiation was delivered with the gantry directly facing the tumor, and the irradiation beam was controlled by a collimator with an appropriate opening matching the tumor diameter; standard dosimetry was performed to calculate the delivered dose to the center of the tumor. For tumor growth and survival experiments, tumor sizes were measured until the tumor reached 2.0 cm in diameter, which was the endpoint of the experiment. Tumor volumes were calculated as volume (mm^3^) = [width (mm)]^2^ × length (mm)/2. For neutrophilic infiltration-examining experiments, different groups of mice were euthanized at specific time points within 48 h after irradiation as indicated for each individual experiment.

### 2.5. CXCR2 Antagonism

When tumor sizes reached about 7 mm in diameter, mice were administered 100 µL of 10 mM CXCR2 antagonist AZD5069 (MedChemExpress LLC (Princeton, NJ, USA)) via oral gavage [23] twice a day for two days. AZD5069 was dissolved in a vehicle solution consisting of 90% corn oil and 10% DMSO. At around 24 h after the first dose of AZD5069, tumors were irradiated as described in the tumor irradiation section.

### 2.6. Neutrophil Depletion

When tumor sizes reached about 7 mm in diameter, mice were injected intraperitoneally (i.p.) with 200 µg of either anti-mouse Ly6G monoclonal antibody (BioXCell, Cat # BP0075-1) to deplete neutrophils or the isotype control antibody (BioXCell, Cat # BP0089) [7]. At 24 h after antibody injection, tumors were irradiated as described in the tumor irradiation section.

### 2.7. mRNA-Seq and Data Analysis

At 3 h, 12 h, and 24 h post-irradiation, different groups of mice were euthanized and about 50 mg of tumor tissues were harvested and stored in RNAprotect^®^ Tissue Reagent from Qiagen (Germantown, MD, USA). RNeasy^®^ Plus Mini Kit from Qiagen was used to extract the total RNA from the stored tumor tissue following the protocol from the manufacturer. mRNA sequencing was carried out using Next Seq SE-75 High Output V2.5 at UTSW Genomics Core. Transcripts per million (TPM) data were used for gene expression analysis. Gene ontology analysis was carried out using the Enrichr online enrichment tool (https://maayanlab.cloud/Enrichr/) [24] on 9 March 2023. The heat maps were generated using ggplot2 (3.4.0) and R (4.0.2) software. For gene set enrichment analysis (GSEA), R (4.3.1) and R packages fgsea (1.26.0) and data.table (1.14.8) were used, and R pakages ggplot2 (3.4.2) and gridExtra v2.3 were used to generate the corresponding plots. 

### 2.8. Chemokine Multiplex Assay

At 3 h (and/or 6 h), 12 h, 24 h, and 48 h post-irradiation, different groups of mice were euthanized, and tumor tissue was collected and stored at −80 °C. A Bio-Plex cell lysis kit and ReadyPrep™ mini grinders from Bio-Rad (Hercules, CA, USA) were used for the tissue protein extraction following manufacturer protocols. Bio-Plex Pro Mouse Chemokine Assay was carried out using a Luminex 200 System (Luminex Co., Austin, TX, USA) at the UTSW Microarray and Immune Phenotyping Core.

### 2.9. Flow Cytometry Analysis

At 24 h after tumor irradiation, mice were euthanized, and tumors were harvested. Approximately 100 mg of tissue was collected and kept in cold cell culture medium on ice. Tumor tissue was cut into small pieces and digested with 125 U/mL collagenase type IV (Gibco) and 60 U/mL DNase I type IV (Sigma-Aldrich, St. Louis, MO, USA) for 60 min at 37 °C [7]. The digested tumor was filtered with 70 µm cell strainers to remove the undigested tissue, and the filtered single-cell suspension was centrifuged and washed with phosphate-buffered saline (PBS). Tumor single cells were first mixed with anti-mouse CD16/32 (BD Biosciences, clone 2.4G2) on ice for 5 min to block Fc receptors. Then, cells were mixed with antibody cocktails containing fluorophore-conjugated antibodies targeting specific cell-surface markers and incubated on ice for 30 min in the dark [25]. Cells were washed with staining buffer (eBioscience™) and fixed with IC Fixation Buffer (eBioscience™). Fixed cells were kept on ice until analyzed by LSR II or LSRFortessa Flow cytometers (BD Biosciences), and the data were further analyzed by FlowJo Software (v10.9.0) (BD Biosciences).

### 2.10. Western Blot Analysis

At specific times after tumor irradiation as indicated in each individual experiment, mice were euthanized, and tumors were harvested. A portion of tumor tissue was placed on dry ice and stored at −80 °C until further analysis. The FastPre-24^TM^ 5G bead beating grinder and lysis system (MP Biomedicals) was used to homogenize the tumor tissue. RIPA buffer (EMD Millipore Corp, Burlington, MA, USA) added with protease (Sigma, Burlington, MA, USA) and phosphatase inhibitors (Sigma) and PMSF was used as tissue lysis buffer. The Bio-Rad DC Protein Assay was used for measuring the protein concentrations. Four percent to 20% gradient precast polyacrylamide gels from Bio-Rad were used for protein electrophoresis. Samples were run at 100 volts for about 1.5 h and transferred at 350 mA for about 1 h following the instruction manual from Bio-Rad Mini Trans-Blot Electrophoretic Transfer Cell. Membranes were blocked with 5% dry milk in Tris-buffered saline with 1% Tween at room temperature for 1 h before primary antibody incubation at 4 °C overnight. After primary antibody incubation, membranes were incubated with HRP-linked secondary antibody at room temperature for 2 h. Pierce^TM^ enhanced chemiluminescent western blotting substrate was used to produce the chemiluminescent signals on the membranes, and X-ray films (Lightlabs, Aurora, CO, USA) were used to detect the signals. ImageJ software (1.53e) [26] was used for band color intensity analysis.

### 2.11. Statistical Analysis

For comparison of tumor-infiltrating immune cells among different groups (*n* > 2), we used one-way ANOVA and Bonferroni’s test to determine the statistical significance. To compare the immune cell and CXCR2+ cell percentages between tumor-bearing mice and control mice, we used Student’s *t*-test for statistical analyses. To determine the statistical significance of gene and protein expression changes of chemokines and other molecules at different time points after RT, one-way ANOVA and Dunnett’s multiple comparisons test were applied. For mice survival curve comparison, the Log-rank (Mantel–Cox) test was used for statistical analyses. All the above-mentioned statistical analyses were conducted with GraphPad Prism (v9.5.0) software.

For tumor growth experiments, the generalized estimating equation (GEE) approach was used to investigate if there was a significant difference in tumor volume over time among the four treatment groups. GEE analysis showed a significant difference in tumor volume among the four treatment groups (*p* < 0.0001). Pairwise comparisons were conducted to identify which treatment pairs yielded significant tumor volume differences between treatment groups using GEE with Bonferroni corrections.

## 3. Results

### 3.1. CXCR2 Blockade Impedes Radiation-Induced Neutrophilic Infiltration into the Tumor

To determine if chemokine receptor CXCR2 is involved in RT-induced neutrophilic infiltration into the tumor, we blocked the ability of CXCR2 to bind to its corresponding ligands by treating mice with CXCR2 antagonist AZD5069 (Figure 1A) [23]. Twenty-four hours after focused RT, tumors were analyzed using flow cytometry with appropriate surface markers and gating to quantitate immune infiltrates (Figure 1B). In the mouse prostate cancer RM-9 model, CXCR2 blockade abolished the RT-induced increases of neutrophil in the tumor (Figure 1C). Contrastingly, CXCR2 blockade did not affect other tumor-infiltrating myeloid populations (Figure 1C). To verify if these findings are consistent in other cancer models, we carried out similar experiments in a mouse colon cancer MC-38 model (Appendix A). Consistent with the findings in the RM-9 model, CXCR2 blockade selectively reduced RT-induced increase of neutrophils without significantly influencing other myeloid populations in MC-38 tumor grafts (Figure 1D). These results demonstrate that CXCR2 is responsible for the recruitment of neutrophils into the tumor upon radiation.

### 3.2. CXCR2 Blockade Has Similar Effects on RT-Induced Tumor Inhibition and Host Survival as Neutrophil Depletion

We previously demonstrated that RT-induced neutrophils in the tumor primarily play an anti-tumor role in the RM-9 tumor model [7]. Since CXCR2 blockade significantly decreases RT-induced neutrophils in the tumor, we next sought to determine if this reduction of neutrophilic infiltration translates into diminished anti-tumor effects of radiation. We treated the RM-9 tumor-bearing mice with CXCR2 antagonist AZD5069 before and after radiation and monitored the tumor sizes (Figure 2A). CXCR2 blockade significantly reduced the tumor growth delay induced by radiation (Figure 2B). Consistent with the tumor growth results, CXCR2 blockade also shortened RT-induced survival (Figure 2C). To confirm the anti-tumor effects of RT-induced neutrophils and to compare the CXCR2 blockade effects with neutrophilic depletion, in a separate experiment, we depleted neutrophils by injecting anti-Ly6G antibody one day before tumor radiation (Figure 2D). Neutrophil depletion was confirmed by measuring the neutrophils in the peripheral blood of these mice (Appendix A). Consistent with our previously reported findings [7], depletion of neutrophils reduced RT-induced tumor inhibition (Figure 2E) and shortened RT-induced survival (Figure 2F). Together with the findings from the former section, these results indicate that CXCR2 blockade not only diminishes RT-induced influx of neutrophils into the TME but also reduces RT-induced effects on tumor growth and survival.

### 3.3. CXCR2 Is Highly and Preferentially Expressed by Neutrophils

As we demonstrated earlier, despite the evident inhibitory effects on RT-induced neutrophil infiltration, CXCR2 blockade did not significantly affect other tumor-infiltrating myeloid cell populations (Figure 1C,D). Although other immune cells such as monocytes and macrophages can also express CXCR2 [27], it is not clear if these cells express similar levels of CXCR2 as neutrophils. The disparate expression levels of CXCR2 between neutrophils and other myeloid cell populations could explain why CXCR2 blockade selectively affected neutrophil migration. We also sought to investigate if lymphoid cells express CXCR2 in tumor-bearing mice. Thus, we used flow cytometry to examine the CXCR2 expression on both myeloid and lymphoid populations in the spleens from control and RM-9 tumor-bearing mice (Appendix A). Unsurprisingly, tumor-bearing greatly increased the percentages of neutrophils, monocytes, and dendritic cells but moderately decreased those of CD4^+^ T cells, CD4 ^+^ CD25^+^ T cells, CD8^+^ T cells, B cells, and NK cells in the spleen (Figure 3A,C). Macrophage levels were unaffected. Interestingly, tumor-bearing increased CXCR2^+^ cell percentages in most types of immune cells examined (Figure 3B,D). Strikingly, almost all neutrophils expressed CXCR2 in tumor-bearing mice, whereas only small percentages of monocytes (3%), macrophages (19%), and dendritic cells (22%) expressed CXCR2 (Figure 3B). Although tumor growth increased CXCR2 expression on lymphoid cells, the percentages of CXCR2^+^ cells were below 1% in these populations (Figure 3D), indicating that CXCR2 blockade would not have significant effects on lymphoid populations. Additionally, we examined CXCR2 expression on myeloid cells in the TME (Appendix A) and found that tumor-infiltrating neutrophils also highly expressed CXCR2 compared to other cell populations (Figure 3E,F). Similar results were found in MC-38 tumor-bearing mice (Appendix A–F). Taken together, our data strongly support that CXCR2 is highly and preferentially expressed on neutrophils, which explains the neutrophilic infiltration into the tumor after RT and the reduction thereof upon the CXCR2 blockade.

### 3.4. Radiation Increases CXCR2 Ligands in Tumor Microenvironment

To further evaluate our hypothesis that the CXCLs/CXCR2 axis mediates RT-induced neutrophilic infiltration, we examined if radiation induces CXCLs in the TME. Five chemokines have been identified to bind with the chemokine receptor CXCR2 in mice, and they are CXCL1, CXCL2, CXCL3, CXCL5, and pro-platelet basic protein (PPBP), also known as CXCL7 [28]. We collected tumor tissue at early time points after tumor irradiation (Figure 4A and Appendix A) and measured mRNA and protein expression of these chemokines by mRNA sequencing and the multiplex chemokine assay, respectively. In the RM-9 tumor model, radiation increased mRNA levels of CXCL2, CXCL3, and CXCL7 at 24 h post-treatment (Figure 4B) and the protein levels of CXCL1 and CXCL2 at 24 h and 48 h post-treatment (Figure 4C). In MC-38 tumor grafts, using the same experimental set-up as in RM-9, radiation increased mRNA levels of CXCL1, CXCL2, CXCL3, and CXCL5 at 24 h post-treatment (Figure 4D) and protein levels of CXCL5 at 24 h and 48 h post-treatment (Figure 4E). These results demonstrated that radiation upregulated gene and protein expression of CXCR2 ligands in the TME early after treatment, supporting our hypothesis that the early activation of the CXCLs/CXCR2 axis by RT mediates RT-induced neutrophilic infiltration into the tumor.

### 3.5. Radiation Increases the Gene Expression of DNA Sensing Pathways in the Tumor Microenvironment in cGAS-Positive MC-38 Model, but Not cGAS-Negative RM-9 Model

To explore the upstream mechanism of RT-induced upregulation of CXCLs, we examined the gene expression of the key molecules involved in cyclic GMP-AMP synthase (cGAS) and absent in melanoma 2 (AIM2) DNA-sensing pathways. Unexpectedly, in the RM-9 model, RT did not significantly affect the gene expression of either DNA-sensing pathways (Figure 5A,B), suggesting neither of these two pathways were activated upon radiation. Contrastingly, in the MC-38 model, we found that RT increased the gene expression of cGAS, STING, TBK1, IRF3, and IFN-β as early as 3 h post-treatment (Figure 5C), suggesting the activation of the cGAS DNA-sensing pathway. Additionally, RT increased the gene expression of AIM2, Caspase 1, and IL-1β within 24 h post-treatment (Figure 5D), indicating the activation of the AIM2 DNA-sensing pathway. Since both IFN-β [29] and IL-1β [16] can trigger the gene expression of CXCLs, the cGAS and AIM2 DNA-sensing pathways might both be involved in the upregulation of CXCLs in the MC-38 model. One possible cause for this discrepancy between the two tumor models is the fact that MC-38 tumor cells express cGAS but RM-9 tumor cells do not express it (Appendix A, Appendix A). Taken together, the results indicated that DNA-sensing pathways might be an upstream mechanism underlying the RT-induced CXCLs in the MC-38 model but not the RM-9 model. The results also suggest that different upstream signaling pathways are involved in RT-induced CXCLs in different types of tumors.

### 3.6. Radiation Increases IL-1β Expression in Tumor Microenvironment

Since IL-1β can upregulate CXCLs [16,17,18], we sought to examine if RT increases the protein level of IL-1β in the TME at early time points. In the RM-9 tumor model, western blot results showed that IL-1β protein expression was upregulated in the tumor as early as 3 h and further increased at 24 h post-radiation (Figure 5E). Similarly, radiation also induced IL-1β expression in the TME as early as 3 h, and IL-1β remained elevated at 24 h post-radiation in MC-38 tumors (Figure 5F). The original blots and densitometry readings are in the Appendix A. These data demonstrate that radiation induced IL-1β expression in the TME at an earlier time point and might be an upstream signal contributing to the upregulation of CXCLs.

### 3.7. Gene Ontology Analysis Shows Gene Enrichment in Related Biological Processes

To further examine our hypotheses, we performed gene ontology analysis on transcriptome data from the mRNA-Seq experiments. We found that RT increased the gene expression for Cellular Response to DNA Damage Stimulus (GO:0006974) at 3 h in RM-9 tumors (Figure 6A) and had a similar and prolonged effect in MC-38 tumors (Figure 6B). RT upregulated the gene expression for Cellular Response to Interleukin-1 (IL-1) (GO: 0071347) at 12 h and 24 h in MC-38 tumors (Figure 6D) and had a similar gene upregulation at 24 h in RM-9 tumors (Figure 6C). Moreover, RT increased the gene expression for Neutrophil Chemotaxis (GO: 0030593) at 24 h in both tumors (Figure 6E,F). TPM values of each individual genes and *p*-values when comparing RT groups at different time points post-RT with the control group are in the Appendix A. In addition, GSEA was performed using the log fold change of TPM values between RT groups at different time points post-RT and the control group (Appendix A), and the results are consistent with those from the heatmaps. These results indicate at least three effects of RT on the TME at early time points: (1) induction of DNA damage; (2) amplified IL-1 production and/or activation; (3) increased neutrophil infiltration. These data corroborate the other findings in this study and support our hypothetical mechanism underlying RT-induced neutrophilic infiltration as depicted in Figure 7.

## 4. Discussion

This study investigated the early events and underlying mechanisms of RT-induced innate immune response, inflammation, and immune modulation in the TME. Previously, it has been demonstrated that RT-induced DNA damage is translated into immune-modulatory signal via multiple DNA-sensing pathways [11,12,13,30,31,32,33]. It has also been demonstrated that neutrophilic infiltration into the TME occurs as an early inflammatory response within 24 h post-RT [7]. However, the cascade of events that is responsible for the neutrophilic infiltration from the DNA-sensing pathway activation has largely been unknown. Here, using two different syngeneic murine tumor models, we demonstrated that pharmacological blockade of CXCR2 significantly reduced RT-induced neutrophilic infiltration, indicating the critical role of CXCR2 in these processes. We also demonstrated that neutrophils highly and preferentially express CXCR2 compared to other immune cells, which provided a mechanistic explanation for why CXCR2 blockade did not affect the numbers of other myeloid cells in the TME upon radiation. Furthermore, we examined the expression of CXCLs in the TME at different time points within 24 h post-RT. The finding of increased levels of CXCLs supports our hypothesis that CXCLs/CXCR2 mediate RT-induced neutrophilic infiltration. To explore upstream mechanisms underlying the increased CXCLs, we found that radiation upregulates gene expression of both the cGAS and AIM2 DNA-sensing pathways in the MC-38 model but not in the RM-9 model, suggesting that multiple and different upstream mechanisms might be involved in the upregulation of CXCLs in different tumor models. The elevated level of IL-1β in the TME within 3 h post-RT suggests the possibility that IL-1β is an upstream signaling molecule for the upregulation of CXCLs. Gene ontology analysis of mRNA-Seq data shows an enrichment in the genes of related biological processes and supports our hypothesis. The cascade of events starting from RT that leads to neutrophilic infiltration is illustrated in Figure 7.

The current study was focused on the mechanism underlying a single high dose (15 Gy) of RT, which induced an acute and transient neutrophilic infiltration to the tumor at 24 h post-RT. Fractioned RT and the delayed effects of RT were not investigated in this study. Our previous study showed that neutrophil numbers returned to baseline levels at 4 days post-RT in the current experimental settings for the RM-9 tumor model [7]. It was reported that RT induced neutrophils at Day 7 after the initiation of fractionated RT with a total of 15 Gy over 14 days [34], and another study showed that neutrophils increased in the tumor at 10 days after the initiation of a daily dose of 3 Gy for 5 days [35]. However, with a single ablative dose of 30 Gy, RT reduced CD11b^+^Gr-1^+^ cells (mostly neutrophils) at day 14 after the treatment [36]. Therefore, RT dosing regimen and sample collection time points could significantly affect the results from neutrophil analysis in the tumor, and these factors should be taken into consideration when interpreting and comparing the data from different studies.

Both gene and protein expression data of the current study showed that radiation increased CXCR2 ligands (CXCLs) in the TME at 24 h and 48 h post-radiation, indicating that these CXCLs participate in RT-induced neutrophilic infiltration. However, there were some differences on the induction patterns of CXCLs between the two tumor models. RT induced gene and protein expression of CXCL5 in the MC-38 model but not in the RM-9 model, whereas RT induced gene expression of CXCL7 in the RM-9 model but not in the MC-38 model. This discrepancy is likely caused by different upstream signaling molecules in these two tumor models and warrants further investigation. Additionally, this specific increase of CXCL5 in the MC-38 model may explain why RT still increased neutrophils, albeit at a lower level when CXCR2 was blocked in this model, whereas RT-induced neutrophilic infiltration was abolished in the RM-9 model. In mice, among the ligands of CXCR2, CXCL5 is the only chemokine that can also bind to CXCR1 expressed on neutrophils [37]. It is therefore possible that the remaining effect of RT on inducing neutrophils in the MC-38 model was mediated by the interaction of CXCL5 and CXCR1.

In the current study, we showed that CXCR2 blockade not only reduced RT-induced neutrophilic infiltration but also reduced RT-induced effects on tumor growth and survival of tumor-bearing mice in the RM-9 model. These results confirm the anti-tumor role of these RT-induced neutrophils in this tumor model, as previously reported [7]. Here, it might be seemingly contradictory to the clinical observations that RT could increase absolute neutrophil count (ANC) [8] and the neutrophil-to-lymphocyte ratio (NLR) [38,39], and these elevated ANC and NLR have a negative impact on survival in patients with cancer. However, these clinical studies were based on the results from fractionated RT with a dosing period of four to nine weeks, which reflected a long-term and often systemic effect of RT on neutrophils. It is unknown if RT can induce neutrophils in the TME or peripheral blood of patients with cancer at 24 h after a single high dose of RT as done with stereotactic radiation.

Our finding that neutrophils highly and preferentially express CXCR2 in both periphery and TME in tumor-bearing mice provides a mechanistic insight on the exclusive blocking effect of the CXCR2 antagonist on neutrophils. This result is in accordance with the previous reports showing higher CXCR2 expression in tumor-infiltrating neutrophils [40] and in the peripheral neutrophils of tumor-bearing mice [41] when compared with other immune cell populations. We showed that tumor-infiltrating neutrophils express a lower level of CXCR2 than peripheral neutrophils, which could be a result of partial CXCR2 internalization after CXCLs/CXCR2 binding [42]. Interestingly, we found that CXCR2 expression levels were higher on myeloid cells compared to lymphoid cells, and some of these myeloid cells are perhaps myeloid-derived suppressor cells (MDSCs). Furthermore, both myeloid cells and their CXCR2 expression increased in the periphery of tumor-bearing mice compared with the control mice. This result suggests that tumor growth, MDSCs increase, and increase in CXCR2 expression could be related and part of the tumor’s immune evasion and suppression strategies, which might be a result of tumor-induced systemic change on cytokine levels that warrant further study.

Radiation has been shown to induce CXCLs in human fibroblasts [43] and cancer cell lines [44,45]. Additionally, RT was reported to induce CXCLs and neutrophil infiltration in the rat liver [46] and mouse intestine [47], and these tissue-infiltrating neutrophils were responsible for RT-induced tissue damage [47]. However, whether irradiation induces CXCLs in the TME has not yet been demonstrated. In the current study, we showed that RT increased CXCLs in the TME at 24 h post-treatment, which provides not only a mechanistic explanation for RT-induced neutrophils but also important insight on how RT modulates the TME by inducing the chemokines.

In this study, we found RT increased IL-1β protein expression at 3 h and 24 h post-RT in both tumor models. Gene ontology analysis showed that RT activated the biological process of the Cellular Response to IL-1. These results suggest that the early upregulation of IL-1β could be an upstream signal leading to increased CXCLs at the later time point of 24 h post-RT. Therefore, IL-1β could be a potential target to modulate neutrophilic infiltration into the tumor upon radiation. Since activation of nuclear factor kappa-light-chain-enhancer of activated B cells (NF-κB) can induce IL-1β gene expression [48] and RT can induce NF-κB activation [49], it is likely that RT-induced upregulation of IL-1β is mediated through NF-κB signaling, warranting further investigation.

Activation of the cGAS DNA-sensing pathway has been demonstrated to increase the production of IL-1β [50]. In addition, activation of the AIM2 DNA-sensing pathway converts pro-IL-1β into the biologically active form of IL-1β and promotes IL-1β secretion. In the current study, we showed RT increased gene expression of both cGAS and AIM2 DNA-sensing pathways in MC-38 tumors. Therefore, the increased level of active IL-1β could be a result of simultaneous activation of the cGAS DNA-sensing pathway, leading to higher gene expression and therefore protein levels of pro-IL-1β, eventually leading to higher levels of active IL-1β by the genes induced by the activation of the AIM2 DNA-sensing pathway (Figure 7B). However, the upstream signaling of RT-induced upregulation of IL-1β in RM-9 tumors has not been identified.

A limitation of the current study is the use of heterotopic subcutaneously transplanted tumor models. In these models, prostate and colon cancer cells were subcutaneously implanted on the legs of mice. Therefore, these non-orthotopic and transplanted models only partially recapitulate the TME in human cancers. The tumor and vasculature growth rates are usually higher in heterotopic transplanted tumor models, whereas spontaneous or orthotopic tumor models have higher MDSC and TGF-β abundance [51]. As a result, as an important player shaping TME, extracellular matrix proteins from cancer-associated fibroblasts [52] might also be significantly different among these tumor models. Therefore, verification of the findings from the current study in spontaneous and orthotopic tumor models is warranted.

## 5. Conclusions

We demonstrated that the CXCLs/CXCR2 axis likely translates the RT-induced DNA-sensing pathway signals leading to the observed RT-induced neutrophilic infiltration into the tumor, which is an early innate inflammatory immune response of RT that leads to further downstream immune-modulatory effects. Mechanistically, RT increases the expression of CXCLs in the TME at 24 h post-RT, and these CXCLs attract neutrophils into the tumor. Compared to other immune cells, neutrophils highly and preferentially express CXCR2, which explains the exclusive blocking effects of the CXCR2 antagonist on neutrophil recruitment into the tumor. RT induced IL-1β protein expression in the TME and increased cellular response to IL-1 at 3 h and 24 h post-RT, respectively, suggesting that IL-1β could be an upstream signal inducing CXCLs. The finding that RT activates both the cGAS and AIM2 DNA-sensing pathways in the MC-38 model but not in the RM-9 model suggests multiple and different upstream signaling pathways are involved in RT-induced upregulation of IL-1β. Therefore, therapies targeting the CXCLs/CXCR2 axis and/or IL-1β signaling may increase the RT-mediated innate response and immune-modulation through shaping tumor-infiltrating neutrophil activity in the TME.

## Figures and Tables

**Figure 1 cancers-15-05686-f001:**
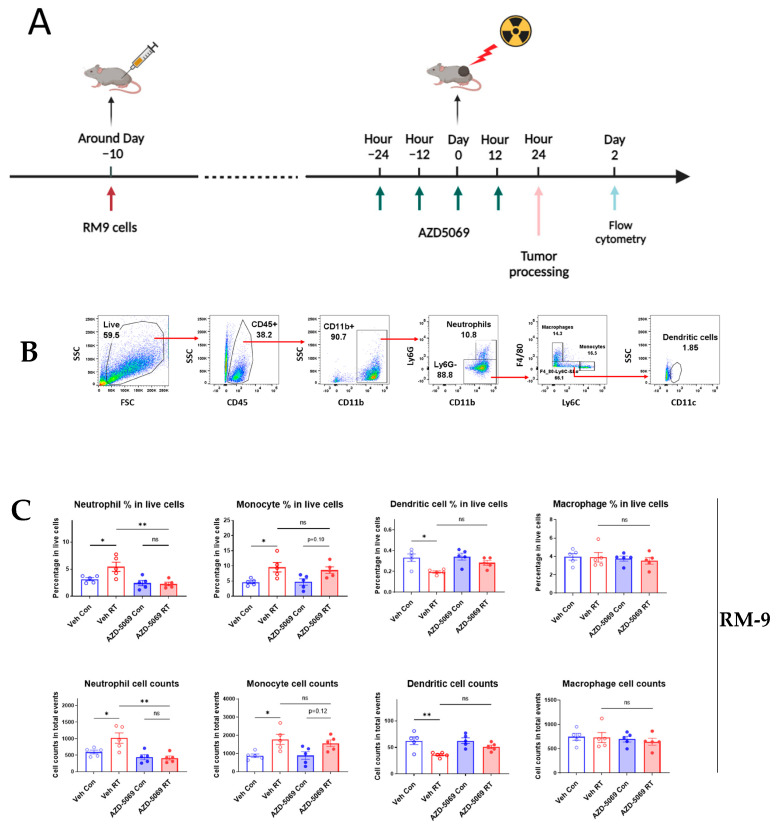
CXCR2 blockade impedes RT-induced neutrophilic infiltration into the RM-9 and MC-38 tumors. (**A**) A schematic illustration of the experimental design. C57BL/6J mice were injected subcutaneously (s.c.) with RM9 tumor cells in the right hind leg. When tumors reached about 7 mm in diameter, mice were randomly assigned to one of four treatment groups and administered AZD-5069 by oral gavage twice per day for two days. One day after AZD-5069 treatment initiation, tumors were irradiated with a single dose of 15 Gy. Mice were euthanized at 24 h after irradiation, and tumors were collected for flow cytometry analysis. (**B**) Gating strategy for tumor-infiltrating myeloid cell populations in the RM-9 tumor model for flow cytometry analysis. Percentages and cell numbers of tumor-infiltrating neutrophils and other myeloid cells in RM-9 tumors (**C**) and MC-38 tumors (**D**). Veh: drug vehicle; Con: sham-treated with RT. Bar graphs showing RT-induced changes on percentages and cell counts of tumor-infiltrating myeloid populations with or without AZD-5069 treatment. n = 5 per group, values = mean ± SEM, * *p* < 0.05, ** *p* < 0.01, *** *p* < 0.001, **** *p* < 0.0001; ns = nonsignificant; one-way ANOVA and Bonferroni’s test were used for statistical analysis.

**Figure 2 cancers-15-05686-f002:**
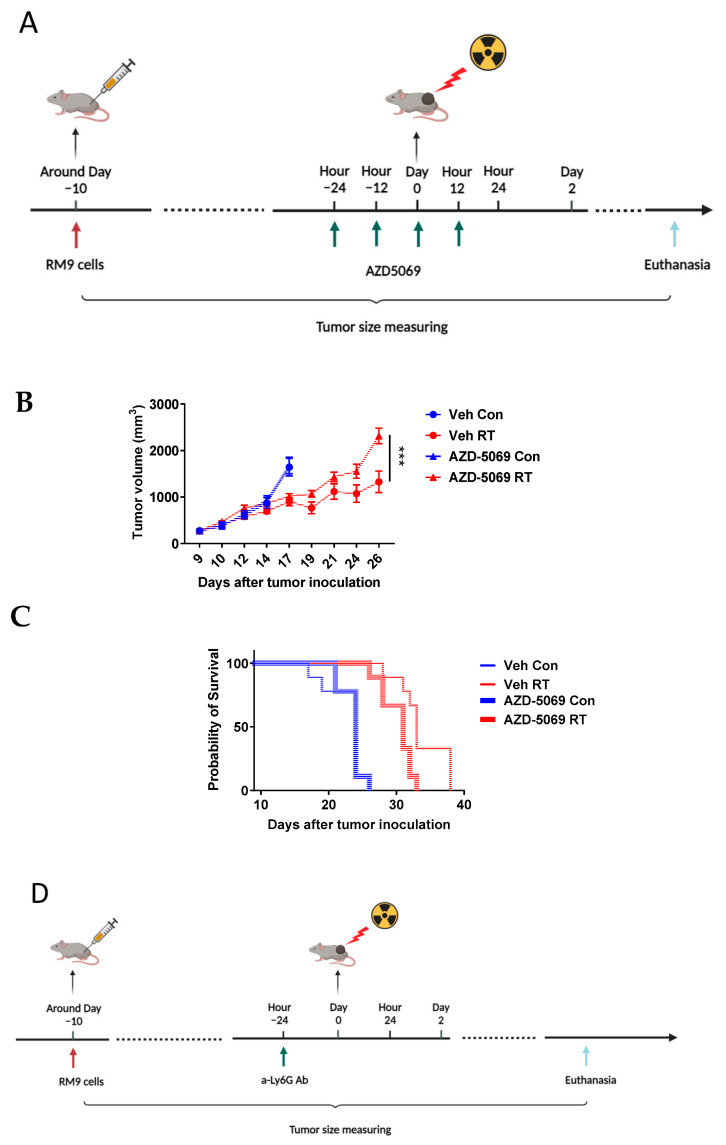
Similar to neutrophil depletion, CXCR2 blockade reduces RT-induced tumor inhibition and decreases survival in the RM-9 model. (**A**) A schematic illustration of the experimental design for the CXCR2 blockade experiment. (**B**) Tumor growth curves with or without CXCR2 blockade. n = 9–10 per group, values = mean ± SEM, *** *p* < 0.001. (**C**) Mouse survival curves with or without CXCR2 blockade. CXCR2 blockade significantly (*p* < 0.05) reduced the survival of mice with irradiated tumors. n = 9 per group. (**D**) A schematic illustration of the experimental design for the neutrophil depletion experiment. (**E**) Tumor growth curves with or without neutrophil depletion. n = 9–10 per group, values = mean ± SEM, *** *p* < 0.001. (**F**) Mouse survival curves with or without neutrophil depletion. Neutrophil depletion significantly (*p* < 0.05) reduced the survival of mice with irradiated tumors. n = 8–9 per group. For tumor growth curves comparison, a generalized estimating equation (GEE) was used for statistical analysis. For survival curves comparison (**C**,**F**), Log-rank (Mantel–Cox) test was used for statistical analysis. Veh: drug vehicle; Con: sham-treated with RT.

**Figure 3 cancers-15-05686-f003:**
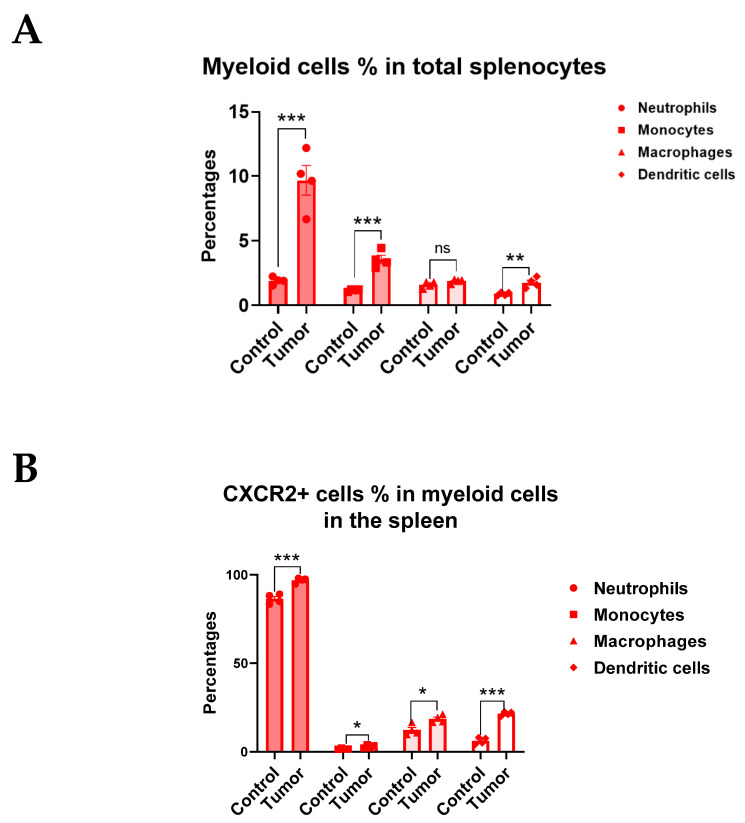
CXCR2 is highly and preferentially expressed on neutrophils. For graphs (**A**–**D**), spleens from C57BL/6J control and RM-9 tumor-bearing mice were collected and processed for flow cytometry analysis. Gating strategies for flow cytometry analysis are shown in Appendix A. n = 4 for both C57BL/6J control and RM-9 tumor-bearing groups, values = mean ± SEM. Student’s *t*-test was used for comparing the data from control mice with those from tumor-bearing mice. ns = nonsignificant, * *p* < 0.05, ** *p* < 0.01, *** *p* < 0.001. (**A**) Bar graphs showing percentages of different myeloid populations accounting for total leukocytes in the spleen. (**B**) Bar graphs showing percentages of CXCR2^+^ cells in different myeloid populations. (**C**) Bar graphs showing percentages of different lymphoid populations accounting for total splenocytes in the spleen. (**D**) Bar graphs showing percentages of CXCR2^+^ cells in different lymphoid populations. For graphs (**E**,**F**), RM-9 tumor tissue was collected and processed for flow cytometry analysis, with gating strategies shown in Appendix A. n = 5, values = mean ± SEM. (**E**) Bar graphs showing percentages of different myeloid populations accounting for total tumor-infiltrating leukocytes. (**F**) Bar graphs showing percentages of CXCR2^+^ cells in different tumor-infiltrating myeloid populations.

**Figure 4 cancers-15-05686-f004:**
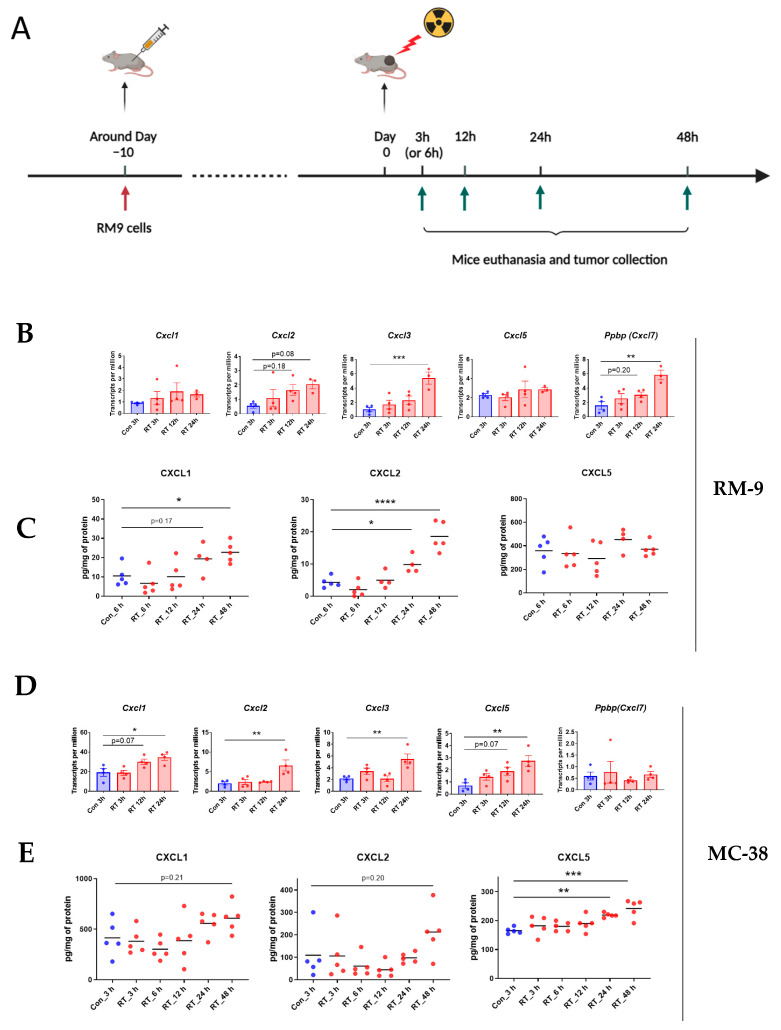
Radiation increases the expression of CXCR2 ligands within 24 h. (**A**) A schematic illustration of the experimental design. Five groups of mice were injected s.c. with RM9 tumor cells on the right hind leg. When tumor sizes reached about 8 mm in diameter, tumors were irradiated with a single dose of 15 Gy. At 3 h (or 6 h), 12 h, 24 h, and 48 h post-RT, different groups of mice were euthanized, and tumors were collected and stored at −80 °C for mRNA sequencing and chemokine multiplex assay. (**B**) In the RM-9 model, mRNA expression of CXCL1, CXCL2, CXCL3, CXCL5, and PPBP (CXCL7) determined by TPM values from mRNA-Seq analysis of the tumor tissue at 3 h, 12 h, 24 h post-RT. n = 3–4 per group, values = mean ± SEM. (**C**) In the RM-9 model, protein expression of CXCL1, CXCL2, and CXCL5 determined by chemokine multiplex assay of the tumor tissue at 6 h, 12 h, 24 h, 48 h post-RT. n = 4–5 per group, values = mean. (**D**) In the MC-38 model, mRNA expression of chemokines CXCL1, CXCL2, CXCL3, CXCL5, and PPBP (CXCL7) in the tumor tissue at 3h, 12h, 24h post-RT. n = 4 per group, values = mean ± SEM. (**E**) In the MC-38 model, protein expression of CXCL1, CXCL2, and CXCL5 in the tumor tissue at 3 h, 6 h, 12 h, 24 h, 48 h post-RT. n = 5 per group, values = mean. (**B**–**E**) * *p* < 0.05, ** *p* < 0.01, *** *p* < 0.001, **** *p* < 0.0001; one-way ANOVA and Dunnett’s multiple comparisons test were used for statistical analyses.

**Figure 5 cancers-15-05686-f005:**
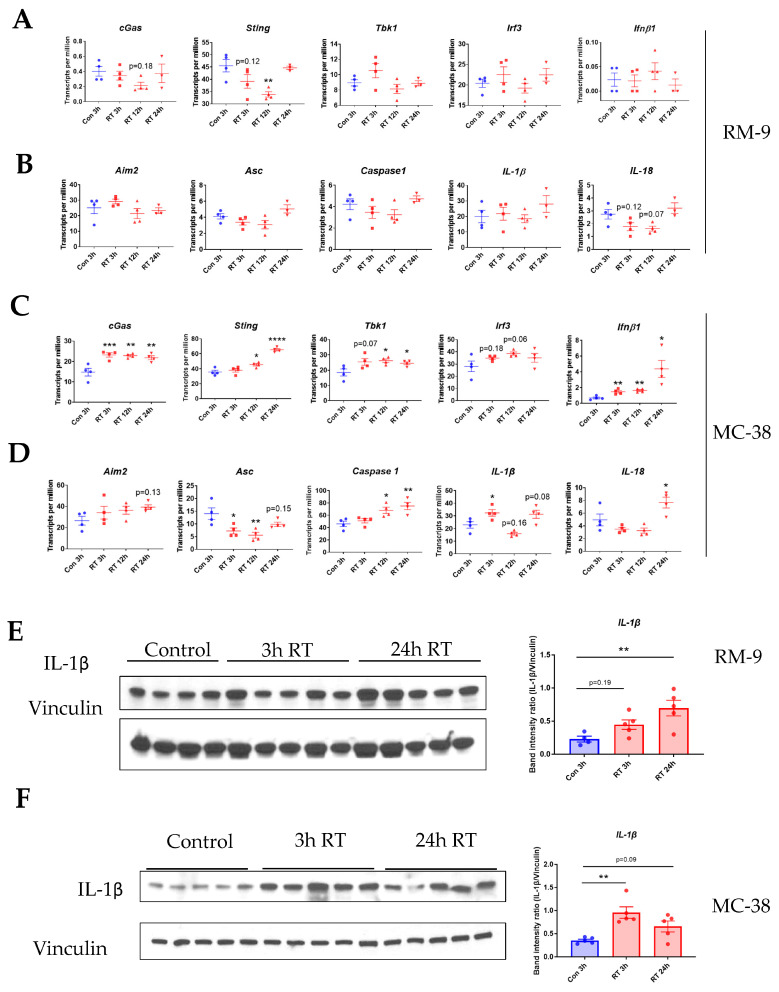
Radiation increases the gene expression of DNA-sensing pathways in the tumor microenvironment within 24 h in the MC-38 model but not in the RM-9 model. The experiment design was the same as described in Figure 4A. In brief, four groups of mice were injected s.c. with RM-9 (or MC-38) tumor cells on the right hind leg. When tumor sizes reached about 8 mm in diameter, tumors were irradiated with a single dose of 15 Gy. At 3 h, 12 h, and 24 h post-RT, different groups of mice were euthanized, and tumor tissues were collected and stored at −80 °C for mRNA-Seq and western blot analysis. (**A**,**B**) In the RM-9 model, mRNA expression of cGAS and AIM2 DNA-sensing pathways, respectively, determined by TPM values from mRNA-Seq analysis of the tumor tissue at 3 h, 12 h, and 24 h post-RT. (**C**,**D**) In the MC-38 model, mRNA expression of cGAS and AIM2 DNA-sensing pathways, respectively, determined by TPM values from mRNA-Seq analysis of tumor tissue at 3 h, 12 h, and 24 h post-RT. (**E**,**F**) In the RM-9 and MC-38 models, respectively, western blot band pictures and intensity analysis of IL-1β and vinculin from tumor tissue samples of control, 3 h, and 24 h post-RT groups. Vinculin was used as the loading control. (**A**–**F**) n = 3–5 per group, value = mean ± SEM, * *p* < 0.05, ** *p* < 0.01, *** *p* < 0.001, **** *p* < 0.0001; one-way ANOVA and Dunnett’s multiple comparisons test were used for statistical analyses.

**Figure 6 cancers-15-05686-f006:**
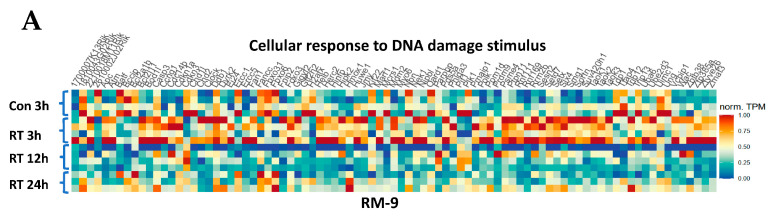
Gene expression heatmaps of activated biological processes from gene ontology analysis. The experiment design was the same as described in Figure 4A. In brief, four groups of mice were injected s.c. with RM-9 (or MC-38) tumor cells on the right hind leg. When tumor sizes reached about 8 mm in diameter, tumors were irradiated with a single dose of 15 Gy. At 3 h, 12 h, and 24 h post-RT, different groups of mice were euthanized, and tumor tissues were collected for RNA extraction and mRNA-Seq analysis. TPM values were used for gene ontology analysis. (**A**,**B**) Heatmaps of normalized individual gene expression involved in Cellular Response to DNA Damage Stimulus (GO: 0006974). (**C**,**D**) Heatmaps of normalized gene expression involved in Cellular Response to Interleukin-1 (GO: 0071347). (**E**,**F**) Heatmaps of normalized gene expression involved in Neutrophil Chemotaxis (GO: 0030593). (**A**,**C**,**E**) are results from the RM-9 model, and (**B**,**D**,**F**) are results from the MC-38 model.

**Figure 7 cancers-15-05686-f007:**
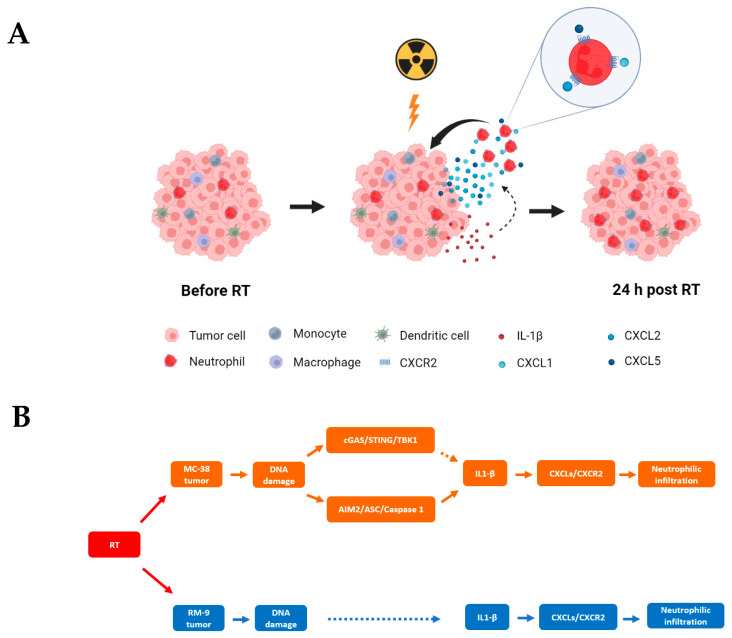
(**A**) A schematic illustration of CXCLs/CXCR2 axis-mediated neutrophilic infiltration into the tumor within 24 h post-RT. RT induces IL-1β in the tumor microenvironment, which can further increase the levels of chemokines CXCL1, CXCL2, and CXCL5. These elevated levels of chemokines can be sequestered by endothelial cells of the blood vessel and form a chemoattractant gradient for neutrophils. The interaction between these chemokines and CXCR2 expressed on neutrophils can promote chemotaxis of neutrophils into the tumor tissue and facilitates the extravasation of neutrophils, which can eventually lead to the increased number of neutrophils in the tumor. (**B**) The hypothetical upstream mechanism underlying the activation of CXCLs/CXCR2 axis upon RT. In MC-38 tumors, the DNA damage induced by RT likely activates (1) the cGAS/STING DNA-sensing pathway which can lead to the upregulation of pro-IL-1β, and (2) the AIM2 DNA-sensing pathway, which can activate IL-1β. In RM-9 tumors, RT induces DNA damage, which likely leads to the increased expression of IL-1β by an as-yet unidentified mechanism. Increased expression and/or activation of IL-1β can upregulate the production of CXCLs, which attract neutrophils into the tumor.

## Data Availability

Data generated in this study are available within the article and its Appendix A or from the corresponding author upon reasonable request. The mRNA-Seq data are publicly available at Gene Expression Omnibus, and the accession number is GSE242237.

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
