# Peer review of "Radiation-Induced Innate Neutrophil Response in Tumor Is Mediated by the CXCLs/CXCR2 Axis"

_cancers, 2023, doi:10.3390/cancers15235686_

Round 1

Reviewer 1 Report

Comments and Suggestions for Authors

The authors examine how irradiation (RT) is impacting the early immune response in murine tumors upon RT and subsequent tumor remission. This laboratory has previously used the same RM-9 prostate grafting model and showed already that RT induces neutrophils (N) to infiltrate the tumor and that ROS produced by these cells are important for tumor remission. The N function could be even further enhanced by treatment with G-CSF prior to RT which is supposed to facilitate N attraction (Takeshima et al., (PNAS 2016, 113, 11300). The previous work has been cited in the current study however is not discussed in light of the new results.

In the current study the authors again used the RM-9 plus a second syngeneic model, MC-38 coloncarcinoma, and confirm that RT (15 Gy) is inducing N infiltration 24 hours after RT. This is mediated through CXCR2 as an inhibitor (AZD5069) is abolishing this N infiltration thereby causing bigger tumor growth and shortened survival of tumor bearing mice. Similar to Takeshima et al., (2016) the authors use anti-Ly6G to deplete N to show that N are instrumental for the tumor reducing effect of RT. Please, could the authors explain how the results shown in Fig. 2E differ to those shown in Takeshima et al., (2016), Fig. 2.

Results from gene expression analysis and confirmation of some secreted candidates by other methods are shown suggesting that the identified candidates could play a role in N activation and attraction that is seen after RT. It is further proposed that some of these candidates play a role in RT-induced DNA damage triggering expression of chemokines/cytokines that would then attract N into the tumor.

The study is carefully done, however the representation of the results has to be improved as well as the interpretation of the results should be more careful as many causal links have not been formally proven although they are likely.

Specific criticism:

1.- The labeling of all figures is too small. This applies in particular to the heatmaps where the resolution of the labeling is very poor.

2.- The cartoons in Fig. 1 and 4 describing the experimental schedule are difficult to understand and should be starting with day 1 when the tumor cells get injected.

3.- Results in Fig. 3 are differently displayed as in all other figures. Please, synchronise by displaying the result for each tumor by a dot.

4.- Please, indicate what the acronyms cGAS and AIMS mean.

5.- The tumor-associated neutrophils are abbreviated as TANs, however this abbreviation is not used in  the manuscript. Either use this acronym or delete it. 

6.- Results 3.2.: Do the authors suggest that at the 24 days time point after N depletion the N are still depleted? Please, could you explain.

7.- Line 310: It should be Figure 1C-D.

8.- Line 329: It should be Figures 3E-F.

9.- The authors looked for several myeloid and lymphoid immune subtypes but not for NK cells. Please, could they explain why?

10.- Many results are shown at gene expression level (e.g. Fig. 5A-D), however only a few candidates are confirmed at protein level. This is understandable, but then the interpretation of the results should be more careful, telling that the results indicate an up or downregulation of molecule x which has to be confirmed at protein level in the future. Please, could you clarify what “Transcripts per million” means?

11.- The heatmaps are not very informative. Please, present the expression as dots for each tumor in a graph including statistical analysis. There is only one control condition for the 3 hours time point (no RT). Can these results be compared to the RT results at 12 and 24 hours? I doubt! If the data for all conditions are shown in  the same heatmap then there should be a clear line that the control is valid for the 3 hours time point only. Nevertheless, a comparison between the different time points after RT is possible and should be better presentated qualitatively as suggested above. There is also the question why the control was chosen for 3 hours rather than for 24 hours. Why 3 hours at all?

12.- The authors have identified different gene classes to be deregulated. This should be enforced by a gene set anrichment analysis using published work and by comparison of the gene expression signatures.

13.- My major criticism concerns the chosen models that were grafted subcutaneously. This setup generates an artefact as an extracellular matrix (ECM) rich capsule is formed around the tumor cell colony. Such a capsule undoubtedly has an impact on the infiltration of immune cells and may in particular impact the infiltration of N. Experiments in models with sponatenous tumor formation might provide an answer to this concern. It is clear that the chosen setup easily facilitates tumor growth assessment after RT. However, these models only partially recapitulate what is happening in human cancer. It is well known that the tumor microencironment (TME) and in particular the solid part of it, namely the ECM cannot be recapitulated in this setup, first because of grafting into the skin (prostate and colon would have been better locations, respectively) and second the short period of growth that is not sufficient to mimic the real TME (as documented in Takeshima et al., 2016, Fig. 4A). The tumor ECM plays an active role in generating an immune suppressive TME and in trapping immune subtypes. Maybe the discrepany between tumor model and real human cancers explains differences in results as mentioned by the authors following Line 621. The authors should discuss the limitations of their model and may like to cite doi.org/10.1016/j.matbio.2023.04.002.

14.- In Figure 7 the authors should be more careful with their statements as many links are just correlations that have not even been proven at protein level. The hypothesized pathways might be true however have not been proven by LOF approaches. This hypothetical nature of the summary should be made clear by choosing appropriate words. Please, add the respective chemokines/cytokines for each model that may be involved and be more specific what the effect on the neutrophils will be, only attraction into the tumor, or others such as N activation, secretion of ROS, induction of netosis etc.

15.- The RT dose of 15 Gy has been used. Please, could the authors explain why this dose has been used and how this would correlate to what is used in the human patient.

16.- It is intriguing that N apparently get recruited into the tumor by different mechanisms in the two tumor models. Please, could the authors speculate more why cGAS/AIMS is likely relevant in one but not the other model? Is this potentially linked to the different chemokine/cytokine expression profile of the respective tumors?

Reviewer 2 Report

Comments and Suggestions for Authors

Radiation therapy (RT) can promote the infiltration of neutrophils infiltration into the tumor microenvironment contributing to control the tumor volume. The authors want to identify the events that has led to the recruitment of neutrophils. The results presented support their hypothesis that RT stimulates the cGAS and AIM2 DNA-sensing pathways which increases the production of IL-1β could be an upstream signal leading to increased expression of CXCL5 and CXCL7. They propose that these chemokines attract in TME the neutrophils which express high level of CXCR2.

The manuscript is very well structured and the adequately performed experiments support their conclusion.

Minor comments

1) Lines 133 – 135: The authors are invited to specify the surface area irradiated on the mice. Did it extend beyond the area occupied by the tumor?

2) Line 136: A tumor diameter of 2 cm was used as an endpoint value to assess the impact of inhibiting neutrophil infiltration into the TME, which is very huge for a C57BL/6J mouse of approximately 7 cm length. Why didn't the authors start their study with tumors 3 to 4 mm in diameter and follow their progression up to 8 to 10 mm? Were the centers of the 2 cm tumors necrotic? On the other hand, assays aimed at measuring the population of immune cells, expression of CXCR2 ligands, DNA sensing pathways and mRNA sequencing were carried out in tumors having a diameter of ~8 mm, which is appropriated.

3) Line 147: How the neutrophil depletion was confirmed?

4) Figure 1: The texts could be written with greater size of character.

5) Figure 5F: Replace “Contro” by “Control”.

6) Section 3.2: The authors are asked to show that treating mice with the α-Ly6G antibody reduces the number of neutrophils in the tumor.

7) Figure 2S: The authors are invited to mention in the legend which from myeloid and lymphoid populations of the spleen are shown in the panels A and B.

Reviewer 3 Report

Comments and Suggestions for Authors

In this study, authors suggest that CXCLs/CXCR2 axis mediates RT-induced innate inflammatory response in the tumor environments, which may affect survival. The followings are my comments.

If CXCR2 is blocked, is the effect of Ly6G antibody canceled?

In figure 4 and 5, do the results come from RNA-seq? If so, authors should validate it by qPCR analysis.

In figure 5F, the signal of IL-1b means mature IL-1b or pro-IL-1b?

Round 2

Reviewer 3 Report

Comments and Suggestions for Authors

The authors showed appropriate responses to my comments, and I have no more comments.